# RT-Seg: A Real-Time Semantic Segmentation Network for Side-Scan Sonar Images

**DOI:** 10.3390/s19091985

**Published:** 2019-04-28

**Authors:** Qi Wang, Meihan Wu, Fei Yu, Chen Feng, Kaige Li, Yuemei Zhu, Eric Rigall, Bo He

**Affiliations:** School of Information Science and Engineering, Ocean University of China, Qingdao 266000, China; 18764819653@163.com (Q.W.); wumeihan@stu.ouc.edu.cn (M.W.); yf0327sky@163.com (F.Y.); lkg@stu.ouc.edu.cn (K.L.); zhu201621102@163.com (Y.Z.); 4e3.rigall.eric@gmail.com (E.R.); bhe@ouc.edu.cn (B.H.)

**Keywords:** side-scan sonar (SSS), real-time semantic segmentation, depth-wise separable convolution, patch-wise strategy

## Abstract

Real-time processing of high-resolution sonar images is of great significance for the autonomy and intelligence of autonomous underwater vehicle (AUV) in complex marine environments. In this paper, we propose a real-time semantic segmentation network termed RT-Seg for Side-Scan Sonar (SSS) images. The proposed architecture is based on a novel encoder-decoder structure, in which the encoder blocks utilized Depth-Wise Separable Convolution and a 2-way branch for improving performance, and a corresponding decoder network is implemented to restore the details of the targets, followed by a pixel-wise classification layer. Moreover, we use patch-wise strategy for splitting the high-resolution image into local patches and applying them to network training. The well-trained model is used for testing high-resolution SSS images produced by sonar sensor in an onboard Graphic Processing Unit (GPU). The experimental results show that RT-Seg can greatly reduce the number of parameters and floating point operations compared to other networks. It runs at 25.67 frames per second on an NVIDIA Jetson AGX Xavier on 500*500 inputs with excellent segmentation result. Further insights on the speed and accuracy trade-off are discussed in this paper.

## 1. Introduction

Side-Scan Sonar (SSS) has become one of the main underwater sensors which has a wide array of applications such as oceanographic surveys [1], geological survey [2] and searching for rescue [3]. The detection of topography and seabed target is realized by the SSS acoustic principle, we can regard the result presented as an image, which is the most intuitive way of visualizing information and has greater significance. Sonar sensor can be mounted on AUVs to quickly perform various ocean missions like path planning [4] and navigation [5,6] by the semantic segmentation results of SSS images. Meanwhile, the real-time segmentation and autonomous recognition technology also promote the development of underwater intelligent devices. The SSS imaging principle is shown in Figure 1. By moving the AUV ahead, the seabed is scanned stripe by stripe, each scan line is recorded as one ping.

Due to the complex underwater environment and interference from its own equipment, SSS images have the following characteristics: (1) Intensity inhomogeneity, (2) poor image contrast and (3) serious noises including reverberation, reflection, acoustic loss, scattering, ambient noise and so on. These characteristics greatly influence feature extraction and image recognition. Researchers used various methods to pre-process the SSS images and studied traditional segmentation algorithms [7,8,9,10] to process them. While it achieves certain results, the process is time consuming and the accuracy remains relatively low [11].

Recent advancement in machine learning with the improvement of computing performance encouraged researchers to further study new algorithms [12] for target detection of the SSS image. The emergence of deep learning [13] promoted more and more Deep Convolutional Neural Networks (DCNNs), designed to better extract features for image recognition and segmentation. Since Long et al. [14] proposed a Fully Convolutional Network (FCN), which extended the original CNN structure into pixel-wise prediction without a fully connected layer, many excellent image segmentation networks focusing more on accuracy appeared one after another. But most methods either suffer from a large number of parameters [15], a large number of floating point operations [16,17,18,19], or both [20,21], and most of their datasets are optical or medical images. In these conditions, they become unusable for many mobile or battery-powered applications such as SSS image analysis tasks. To this end, we design an effective network structure to satisfy the real-time semantic segmentation for SSS images.

In this work, we focus on the practical trade-off between speed and accuracy involved in designing segmentation architectures for a specific application and platform. We are inspired by the SegNet [22], an encoder-decoder structure, where the encoder uses the pooling layer to gradually reduce the spatial dimension of the input data, and the decoder gradually recovers the details of the target and the corresponding spatial dimension through a network layer such as a deconvolution layer. This structure has been widely applied to the field of semantic segmentation. One of the main contributions of RT-Seg is its novel encoder block. It utilizes the depth-wise convolution where the parameters are reduced significantly, and the 2-way branch ensure the features of SSS images can be extracted better. Moreover, we also bring some innovations in the decoding method which can recover the target details more accurate. In particular, to deal with the high-resolution SSS image, we use the patch-wise strategy [23] splitting the original image into local patches and sending them to network training. The experiment of real-time semantic segmentation by our network was carried out on our AUV through its GPU module.

The remainder of the paper is organized as follows: In Section 2, we review the related work. Then we give detailed explanation of our RT-Seg structure in Section 3. In Section 4, we introduce the implementation details of our datasets and network training. Section 5 shows the experimental results and analysis in which we evaluate different networks on two SSS images datasets, followed by a conclusion in Section 6.

## 2. Related Work

### 2.1. SSS Images Segmentation

Due to the different sources of noise interference, there is no general segmentation algorithm for SSS images. Although unsupervised methods [8,24,25] like clustering can be used for SSS image segmentation, their results are not necessarily semantic. These methods cannot subdivide the classes they trained to segment, but can perform better at searching for regional boundaries. Early semantic segmentation methods relied on handcrafted features such as Histogram of Oriented Gradient (HOG) [26] and Scale-invariant feature transform (SIFT) [27] combined with common classifiers and hierarchical graphical models. This continues until the revival of deep learning. Convolutional neural networks (CNNs) are robust to face noisy images, they not only help image recognition, but also greatly promote the development of semantic segmentation.

### 2.2. Semantic Segmentation

Semantic segmentation is one of the basic tasks in computer vision, which consists of dividing a visual input into different semantically interpretable categories. “Semantic interpretability” describes classification categories as meaningful in the real world. Moreover, it has numerous benefits in robot-related applications [28,29,30]. Figure 2 is an example of semantic segmentation, aiming to predict the class label for each pixel in the image.

In convolutional neural networks, it is common to periodically insert a pooling layer between successive convolutional layers. Its function is to progressively reduce the spatial size of the representation and extract the main features, leading to a decrease in the number of parameters in the fully connected layer. Through convolution and pooling operations, a number of features is obtained. In the fully connected layer, we will connect all the features and send the output values to the classifier (such as the softmax classifier). Due to the large number of parameters of the full connection layer, and the image structure will be broken. Therefore, the researchers began to use the convolution layer to “replace” the fully connected layer.

In 2014, Long et al. from the University of California in Berkeley, CA, USA, proposed a Full Convolutional Network (FCN), which allows convolutional neural networks to perform dense pixel prediction using convolution layer without the need of the fully connected layer. Using this method, much faster than the image block classification, an image segmentation map of any size can be generated. FCN becomes the cornerstone of deep learning applied to semantic segmentation.

In addition to the fully connected layer, another major problem in semantic segmentation is the pooling layer. Although the pooling layer expands the receptive field, it causes the loss of location information. However, since semantic segmentation requires that the label map fits perfectly, location information needs to be preserved. There are two different structures to solve this problem. The first is the encoder-decoder structure. The encoder gradually reduces the spatial dimension of the pooling layer, and the decoder repairs the details and spatial dimensions of the object. There is a quick connection between the encoder and the decoder, which helps the decoder better repair the details of the target. UNet [31] and SegNet [22] are the most commonly used structures in this approach. Most recently, Lin et al. proposed RefineNet [15], which extended the standard encoder-decoder approach by adding residual units inside the skip-connections between encoder and decoder. The second method uses the dilated convolutional structure to remove the pooling layer, such as Deeplab [17]. The current state-of-the-art network in the popular benchmark dataset PASCAL VOC [32] belongs to DeepLab-v3 [18] (86.9% means intersection-over-union (MIOU) on test sets) and DeepLab-v3 + [33] (89.0%).

To obtain more accurate segmentation boundaries, researchers also attempted to cascade their DCNNs with post-processing steps, such as using conditional random fields (CRF) [16,17].

### 2.3. Real-Time Semantic Segmentation

Recently, real-time semantic segmentation has begun to draw attention. Paszke et al. introduced ENet [34] as an effective lightweight segmentation network. Chaurasia et al. [35] proposed the LinkNet architecture using ResNet18 as the encoder. LinkNet achieves a better accuracy than ENet. However, ENet overcomes it in terms of computational efficiency. Researchers from the University of Adelaide and the University of Melbourne have solved the problem by deploying multitasking models on computing-constrained, achieving the state-of-the art effect. Simultaneously implemented semantic segmentation and depth estimation [36]. Nonetheless, these networks are designed to process common images, when the image size is about 200–800 pixels, and pay little attention to specific applications such as SSS image processing.

## 3. Network Architecture

In this section, we will introduce a detailed description of RT-Seg. Its structure is divided into two main modules: The encoding module is mainly responsible for extracting different levels of features from the SSS image, the corresponding decoding module is responsible for upsampling these features to the original input resolution and calculate the final class probability maps. Our network architecture is shown in Figure 3.

### 3.1. Encoder Architecture

The encoding structure of RT-Seg is built in a sequential way by stacking the initial block and four novel encoder blocks as shown in Figure 3.

The initial single block of our network possesses a convolutional layer with 16 filters which adds up to 16 feature maps after convolution, then max-pooling performs to reduce the input size by half, that is presented in Figure 4a. The encoder block in our proposed architecture is based on the inverted residual block in MobileNet-V2 [37] which is a light-weight network. Our novel encoder block is shown in Figure 4b.

In our encoder block, the 2-way branch is applied to get different scales of receptive fields. One way of the branch uses a 3 × 3 kernel size, the other way uses a 5 × 5 kernel size to learn visual patterns for large objects. Therefore, the RT-Seg is a “wider” architecture that makes efficient use of its minimized number of layers to achieve accurate segmentation in real time.

The first 1 × 1 convolution is the “expansion” layer, in order to increase the number of channels and obtain more features, which is important for next operations. If the latter DW, convolutional layers receive a small number of input channels; they can only extract features in low-dimensional space, limiting its potential performance. To solve this issue, we use the 1 × 1 convolution layer to ensure that the DW convolutional layers are performing in a relatively higher dimension. As shown in Figure 4b, we use t to represent the expansion factor, which is the multiple of the input channel expansion.

The left branch consists of 3 × 3 depth-wise (DW) convolution and the 1 × 1 point-wise (PW) convolution, also known as Depth-Wise Separable Convolution: The DW convolution is responsible for filtering the inputs channels, and the PW convolution is responsible for combining the results of the DW convolution. It separates the calculation of regions and channels into two steps while the standard convolution considers their calculation simultaneously. Many efficient neural networks [38,39] used the Depth-Wise Separable Convolution widely. We can intuitively see the difference between these two types of convolutions in Figure 5.

Moreover, the computational complexity of Depth-Wise Separable Convolution can be greatly reduced. Suppose that the size of our input feature map is H × W × C_in_, the convolutional kernel size is k × k, and the number of filters is C_out_. The calculated amount after standard convolution is:H × W × C_in_ × k × k × C_out_(1)
The amount of calculation after Depth-Wise Separable Convolution is:H × W × C_in_ × k × k + C_in_ × C_out_ × H × W(2)

In our encoder block, when k is set to 3, we can conclude that the calculation of the Depth-Wise Separable Convolution is reduced to about 1/9 of the standard convolution and when k is set to 5, we can conclude that the calculation of the Depth-Wise Separable Convolution is reduced to about 1/25 of the standard convolution.

We placed batch normalization [40] and Rectified Linear Unit (ReLU) after the DW convolution layers. The mathematical expression for the activation function ReLU is as follows:(3)f(x)=max(0,x)
When the input signal *x* is negative, the output is 0; when *x* is positive, the output is equal to the input. We found that ReLU only needs a threshold to get the activation value. In order to ensure the expression ability of the model and prevent information loss, we replace the ReLU with linear activation after the PW convolution layers. Although the activation function can effectively increase nonlinear classification in high-dimensional space, it will destroy features in low-dimensional space. Meanwhile, for the negative input of the ReLU function, the output is all zero, since the original feature has been “compressed”, and then through ReLU, it will inevitably “lose” some features. Therefore, it is not appropriate to use ReLU after dimension reduction.

The right branch in Figure 4b is also a Depth-Wise Separable Convolution, but we adopt a 5 × 5 DW convolution, which will increase the receptive field. The larger the receptive field, the wider range of original input the DW convolution can deal with. Moreover, the high-level feature maps are generated by a larger receptive field, allowing to obtain richer semantic information.

Instead of merging the 2-way branch with an element-wise addition like in inverted residual block, we use shortcut for filter concatenation in our encoder block as a concatenation of channels. This is heavily inspired by [41]. On the one hand, it increases the width of the network rather than the depth. The deeper the network, the easier the backpropagation gradient disappears (vanishing gradient problem) and the more difficult to optimize the model. Balancing the width and depth of the network can improve the performance of deep neural networks. On the other hand, it combines features of different scales, we can more effectively utilize the features and enhance the transfer between features.

The last layer of the encoder block is the max-pooling layer with non-overlapping 2 × 2 windows, which downsamples the feature maps of the convolutional layer by calculating the maximum value of each window. The purpose is to retain key features while reducing parameters and calculations. We only store the max-pooling indices, the locations of the maximum feature value in each pooling window, and use it to upsample the feature maps for recovering spatial information. We will introduce the details in Section 3.2.

The calculation process of our encoder block implementation is shown in Table 1, where t is the expansion factor and k represents the number of input channels. Our experiments show that when t equals 6, we will get the best segmentation performance as shown in Section 5.

### 3.2. Decoder Architecture

As shown in Figure 6, the input of our decoding block consists of two parts: One from the encoder information and the other upsamples its input feature map using the memorized max-pooling indices from the corresponding encoder feature map. The decoder combines the channels of these two parts instead of adding them.

First, we can see the implementation of max-unpooling layer from Figure 7. We use the pooling indices from encoder feature maps to perform upsampling its input feature maps in decoder block. This operation does not involve deconvolution, which greatly speeds up the training time and allows to reduce the memory requirement. Still, it will hurt accuracy. Then we transfer the entire feature maps from encoder blocks to the corresponding decoder blocks and concatenate them to upsampled (using pooling indices) decoder feature maps. This operation can effectively preserve the edge detail information from the original image. Since the decoder can access the knowledge learned by the encoder at each layer, the decoder can use fewer parameters and get better recovery of image feature information. This decoding method can make the overall network more effective compared to existing segmented networks for real-time operations. Furthermore, with little modification, this decoding method can be embedded in any encoder-decoder network architecture.

Our decoder block is illustrated in Figure 6. We combine Xn (n = 1,2,3,4) from Encoder block in Figure 4b and the upsampled max-unpooling layer. Then low-dimensional output is obtained by the convolutional layers of 1 × 1, 3 × 3.

## 4. Implementation Details

In this section, we present the details of implementation of the datasets and network training for SSS images.

The overall experimental process is shown in Figure 8. We obtain the sonar data in XTF format from the sonar sensor mounted on an AUV, these data are stitching together in one single image after being parsed and then saved as a georeferenced grayscale image at high resolution. We collected sonar data from two locations, and then used the patch-wise strategy to divide the high-resolution sonar images into two datasets, one of which contains coral reef images (Dataset-1) from the South China Sea region, and the other one consists of sand wave images (Dataset-2) from Tuandao Bay in Qingdao, China.

### 4.1. Patch-Wise Strategy and Datasets

The high-resolution sonar images are shown in Figure 9a,b, where the average image size is 10,000 × 7200 pixels (after interpolation). The pixel values in the image are determined by the reflected strength of the echo and the distance from objects. The maximum pixel value is 255 as black color and the minimum value is 0 as white color. The black area in the middle of the image is a blind spot called water column, indicating that the sound wave propagates in the water. When the sound wave first reaches the sea floor and returns signal, the black area disappears and the seafloor echo signal begins to appear in the image. The proportion of the black area in the sonar image depends on the height of the sonar sensor from the sea floor. Since it interferes with our SSS image segmentation, we need to remove the water column area from the image as shown in Figure 9c.

Due to the limitations of GPU memory, the networks cannot deal with such large images directly. Resizing images is a common way for network training but causes the loss of image detail information. Therefore, we use patch-wise strategy to solve this problem, which is shown in Figure 9d.

We split the high-resolution sonar image (the water column removed image) into local patches (500 × 500 pixels) with a stride in the order of 50 (a stride of 500 for test set). A certain degree of overlap can reduce image loss caused by splitting and increase the adequacy of the training set. We obtained 20 high-resolution images of coral reefs and 20 high-resolution images of sand waves respectively. After performing the patch-wise strategy, we got two datasets (Dataset-1 and Dataset-2), each of which contains 11,300 images for training and 9670 images for testing. Each image is divided into two classes: target and background. In addition, these two datasets are trained and tested separately in order to verify the generalization performance of our network.

For providing ground truth images to the datasets, we use LabelMe, an open access image annotation tool. It was developed by the Computer Science and Artificial Intelligence Laboratory (CSAIL) of Massachusetts Institute of Technology (MIT) that allows users to perform image annotation manually. The project source code is open. (https://github.com/CSAILVision/LabelMeAnnotationTool). Performance comparison of different networks when the inputs are of size 500*500 is illustrated in Figure 10.

### 4.2. Details of Training

#### 4.2.1. Loss Function

We denote raw input image (h×w) by $X_{n}=\{x_{ij}^{\left(n\right)},i=1,\ldots ,h,j=1,\ldots ,w\}$, the corresponding ground truth for image $X_{n}$ is $Y_{n}=\{y_{ij}^{\left(n\right)},i=1,\ldots ,h,j=1,\ldots ,w\},y_{ij}^{\left(n\right)}\in \{0,1\}$. Our training dataset is $T=\{(X_{n},Y_{n}),n=1,\ldots ,N\}$.

For a typical SSS image, the distribution of background/target is heavily biased, there is large variation in the number of pixels in each class in the training set. To solve this problem, we use the balanced cross entropy (Balanced CE) as the loss function for training networks. We introduce a weighting factor.

First, Cross Entropy (CE) based on binary classification is calculated as follows:(4)$$L_{CE}=-\log(P)$$
(5)$$P=\left\{ \begin{array}{l} p\;\mathrm{if}\;y=1\\ 1-p\;\mathrm{if}\;y=0 \end{array}\right.$$
While y = 1 represents the target and y = 0 the background, p is the predicted probability for the target class, p∈[0,1].

For Balanced CE, we introduce a weighting factor for each class. The weighting factor is:(6)$$\alpha_{w}=\left\{ \begin{array}{l} \alpha \;\mathrm{if}\;y=1\\ 1-\alpha \;\mathrm{if}\;y=0 \end{array}\right.$$
$\alpha_{w}\in [0,1]$, where α = |Y-|/(|Y+|+|Y-|),1−α = |Y+|/(|Y+|+|Y-|). |Y-| and |Y+| denote the background and target ground truth label sets, respectively.
(7)$$L_{BalancedCE}=-\alpha_{w}\log(P)$$

#### 4.2.2. Optimization

We trained our RT-Seg using stochastic gradient descent (SGD) with a batch size of 100 examples, momentum of 0.9, fixed learning rate of 0.001 and weight decay of 0.0005. We found that this small amount of weight decay is important for model learning, which reduces the training error of the model.

The update rule for weight w is:(8)$$v_{i+1}:=m\cdot v_{i}-w_{d}\cdot \alpha \cdot w_{i}-\alpha \cdot {\left\{\frac{\partial L}{\partial w}{|}_{Wi}\right\}}_{Di}$$
(9)$$w_{i+1}:=w_{i}+v_{i}+1$$
where i is the iteration index, m is a hyper-parameter of momentum, $v_{i}$ is the momentum variable, α is the learning rate, $w_{d}$ is the weight decay, ${\left\{\frac{\partial L}{\partial w}{|}_{\mathrm{wi}}\right\}}_{\mathrm{Di}}$ is the average value over the i-th batch Di of the objective derivative of w, evaluated at w_i_. We train the variants until the training loss converges, after then we select the model which performs the highest accuracy on our training datasets.

### 4.3. Evaluation Metrics

Semantic segmentation has four common metrics, introduced below. Let *k_ij_* be the number of pixels of class *i* predicted to belong to class *j*, where there are c different classes, and let $k_{i}=\sum_{j}k_{ij}$ be the total number of pixels of class *i*. The four metrics are calculated as follows:(10)$$PA=\frac{\sum_{i}k_{ii}}{\sum_{i}k_{i}}$$
(11)$$MPA=(1/c)\sum_{i}k_{ii}/k_{i}$$
(12)$$MIoU=(1/c)\sum_{i}\frac{k_{ii}}{\left(k_{i}+\sum_{j}k_{ji}-k_{ii}\right)}$$
(13)$$FWIoU=(1/\sum_{t}k_{t})\sum_{i}\frac{k_{i}k_{ii}}{\left(k_{i}+\sum_{j}k_{ji}-k_{ii}\right)}$$

Pixel Accuracy (PA), which is the simplest metric, computes the ratio of the correctly predicted pixels to the total pixels. Mean Pixel Accuracy (MPA), calculates the proportion of correctly classified pixels in each class, then finds the average for all classes. Mean Intersection-Over-Union (MIOU), a standard measure for semantic segmentation, calculates the ratio of intersection to union between the prediction and ground truth. Frequency Weighted Intersection-Over-Union (FWIOU) is an improved version of MIOU, which sets weights for each class based on their frequency in images.

## 5. Experimental Results and Analysis

In our experiments, all the training networks were performed on NVIDIA Quadro M5000 card using Pytorch framework. We trained the networks for roughly 200 epochs, which took about eight days on two NVIDIA Quadro M5000 GPUs. The well-trained model was then tested on an NVIDIA Jetson AGX Xavier embedded system module. We compare RT-Seg with the existing network architectures on different indicators.

### 5.1. Qualitative Results and Analysis

Figure 10 contains various segmentation results produced by our architecture where the t is set to different values, where purple color represents the background and yellow color represents the target. The parameter t is the channel expansion factor in our encoder block, introduced in Section 3.1.

We can observe that RT-Seg-t-4 gives the coarsest predictions, which means if the number of channels after expansion is small, the encoder part cannot extract features well enough and the network cannot learn sufficiently complex equations to reach the expected performances in applications. These unclear edge problems are progressively corrected with the increase of expansion factor. RT-Seg-t-6 yields superior performance, particularly with its ability to delineate boundaries. When we continue to increase t (e.g., t is 8), on the one hand, it will lead to a large increase of network parameters, influencing the network computational time. On the other hand, simply expanding the number of network channels cannot significantly improve the network accuracy as shown in Table 2 and Table 3. However, from the visual segmentation results, RT-Seg-t-8 shows a stronger adaptability of SSS images in both datasets. Although it means better learning ability, it also become easier to predict what is not needed or wrong as in Figure 10(e3).

It is worth noting that the error of manual labeling will slightly affect the accuracy of network training. For example, in Figure 10(d4), the prediction result looks more like the image corresponding label, rather than the manually labeled ground truth in Figure 10(b4).

### 5.2. Quantitative Results and Analysis

Evaluating the generalization capabilities of the model and dataset demonstrates its utility for applications. Table 2 and Table 3 show the quantitative comparison of different networks for semantic segmentation with class balancing, performed on the test datasets. We can see that UNet provides best results in accuracy on both datasets while it is not computationally time-efficient. Meanwhile, RT-Seg-t-8 is slightly worse than UNet but achieves better performance compared with other networks. Moreover, RT-Seg-t-6 can greatly reduce the number of parameters and floating point operations to realize real-time processing without a significant drop in accuracy compared with RT-Seg-8.

### 5.3. Inference Time

We report the inference time for a single input frame with different resolutions and the number of frames that can be processed per second on the Nvidia Xavier embedded system module in Table 4. As evidenced from Table 4, RT-Seg-t-6 can process 25 frames per second, meeting the real-time requirements and providing high frame rates for real-time applications and allowing the actual use of deep neural network models with encoder-decoder architecture.

Figure 11 shows the trade-off between accuracy and speed at resolution of 500 × 500. The UNet has the advantage of accuracy, but its inference time for a single image input is much longer than other networks as shown in Figure 11a,c. We can also clearly see that the number of frames that RT-Seg-t-6 can be processed per second is much more than for RT-Seg-t-8. Meanwhile, there is no significant accuracy drop for RT-Seg-t-6 as shown in Figure 11b,d. However, during the testing of SSS image datasets, SegNet and ENet have no obvious advantages, neither in terms of accuracy nor speed.

### 5.4. Hardware Requirements

A comparison of computational time and hardware resources required for various network architectures shown in Table 5. RT-Seg-t-6 is superior to other networks in Giga Floating-point Operations Per Second (GFLOP), but it is on par with it in accuracy. RT-Seg-t-6 is very efficient, it can fit the entire network in the embedded processor with relatively few parameters and offer real-time performance on embedded devices, even meet the extremely strict memory limits.

### 5.5. Real-Time Process and Analysis

The experiment of real-time sonar data processing was carried out on our AUV through its GPU module (NVIDIA Jetson AGX Xavier). According to the above analysis, RT-Seg-t-6 achieves a significant trade-off between accuracy and speed, so we use it for real-time test. The experimental process of real-time test is shown in Figure 12.

We will receive 60 pings sonar data per second from the sonar sensor. The time t1 for parsing the raw sonar data is about 0.2 s. When the slant range is set to 120 m (the maximum range), there are 7200 samples per ping, which corresponds to 7200 pixels. We got an SSS image of 60*7200 pixels after data parsing. Then we interpolate it to 500*7200. In total, we will get an SSS image with 500 × 7200 pixels per second. The interpolation time is completed by a CUDA parallel programming, so this time can be ignored. We use a patching strategy to split the high-resolution sonar image into 500*500 size of 15 local patches for 0.1s. In Table 4, we can see that the inference time of RT-Seg-t-6 for a single input of 500*500 pixels is 38.9 ms. The total inference time t3 for 15 images is 0.583 s. The total processing time of sonar data is 0.883 s, which means that the semantic segmentation of sonar data takes less time than the arrival of new sonar data to process.

Therefore, the RT-Seg-t-6 has good generalization performance and can be used in the tasks requiring the real-time segmentation performances on SSS images.

The experimental results show that RT-Seg-t-6 performs better accuracy and faster, but still has similar resources consumption with ENet. Although it requires less GFLOPs in real-time segmentation tasks, its number of parameters and model size remain larger than the ones from ENet. For further study, it would be interesting to study the small model size and number of parameters of ENet and search for adapting them to our network in order to reduce their relative experimental performances. Moreover, due to the noise interference of SSS images, the segmentation results are not ideal. In the future, we will preprocess the SSS images before network training to improve the accuracy, such as median filtering, contrast enhancement or histogram equalization.

## 6. Conclusions

In this work, we have proposed a novel neural network architecture designed specifically for real time semantic segmentation of SSS images. We achieved that by proposing the novel encoder architecture and decoder methods. Experiments show an excellent trade-off between the speed and accuracy of our network. Meanwhile, we use a patch-wise strategy to process high-resolution sonar images which avoids the loss of useful image information. Due to the particularity of the SSS images, our work provides tremendous benefits in this task, while surpassing the existing baseline models, which have an order of magnitude greater computational and memory requirements. RT-Seg-t-6 runs primarily on embedded mobile devices, and NVIDIA Jetson AGX Xavier can be placed in AUVs to process sonar data in real time. We hope that our approach can be applied with any segmentation network of other SSS image datasets in the future.

## Figures and Tables

**Figure 1 sensors-19-01985-f001:**
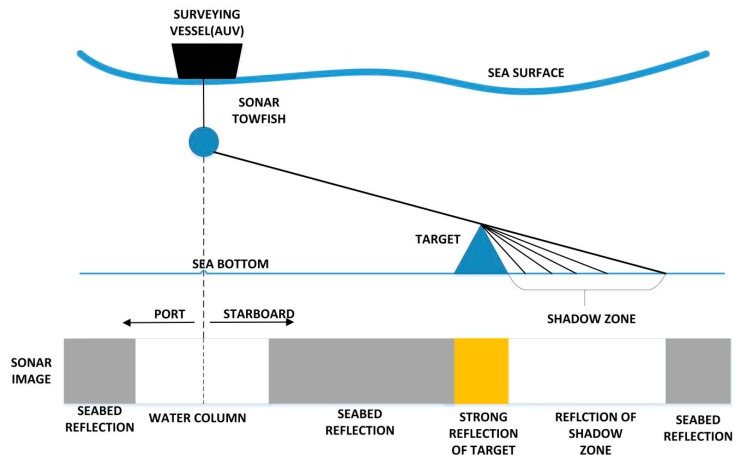
The SSS imaging principle.

**Figure 2 sensors-19-01985-f002:**
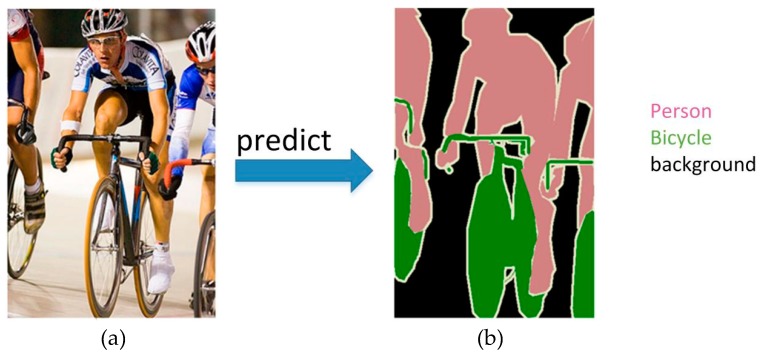
An example of semantic segmentation. (**a**) The test image containing three classes: Person, bicycle and background; (**b**) The well predicted segmentation result.

**Figure 3 sensors-19-01985-f003:**
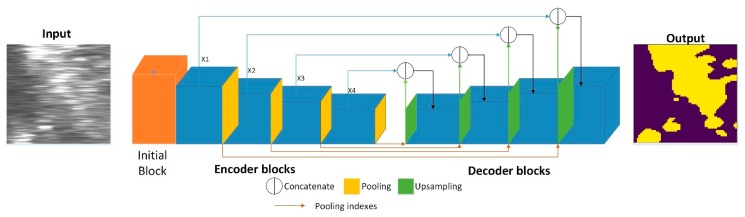
An illustration of the RT-Seg architecture.

**Figure 4 sensors-19-01985-f004:**
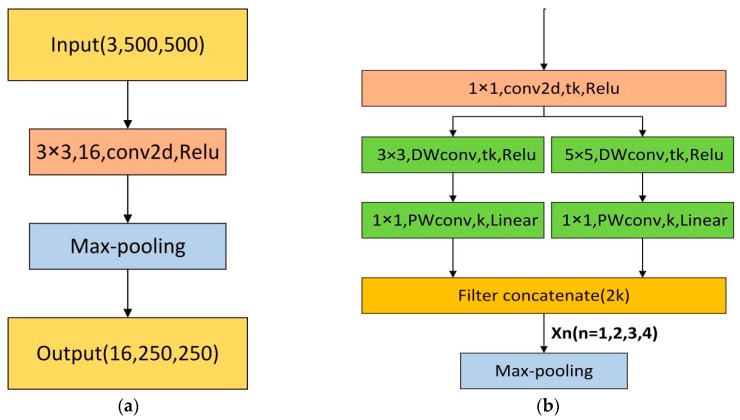
(**a**) Initial block; (**b**) RT-Seg encoder block, where k is the number of input channels, t is the channel expansion factor (t > 1) and n represents the serial number of our encoder blocks, we mark the number of filters at each layer.

**Figure 5 sensors-19-01985-f005:**
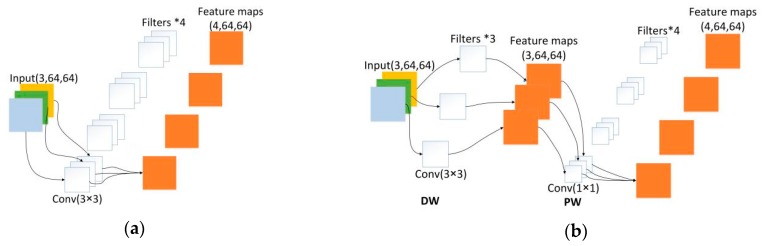
Comparison of two different convolution operations, supposing the input size is 64*64 with 3 channels, and the number of filters is 4. (**a**) The standard convolution; (**b**) Depth-Wise Separable Convolution which consists of DW convolution and PW convolution.

**Figure 6 sensors-19-01985-f006:**
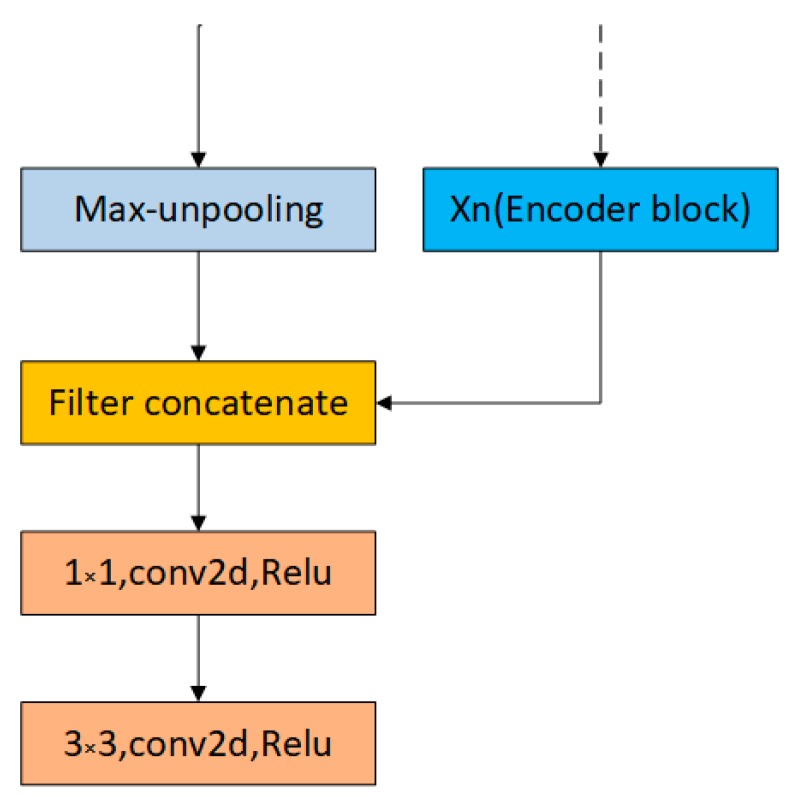
Our decoder block structure.

**Figure 7 sensors-19-01985-f007:**
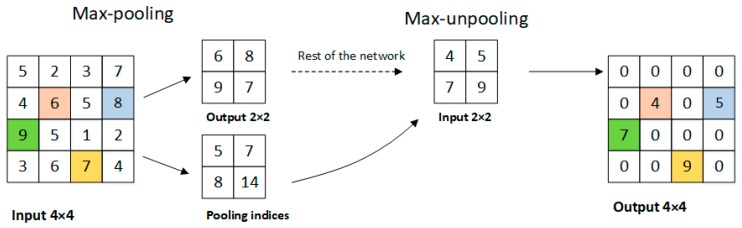
The implementation of the max-unpooling layer.

**Figure 8 sensors-19-01985-f008:**
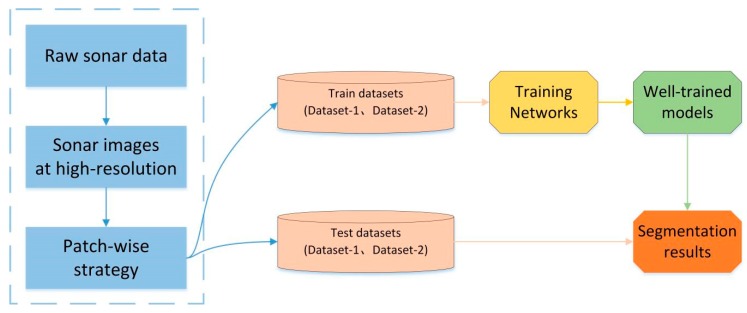
The experimental process for network training and testing.

**Figure 9 sensors-19-01985-f009:**
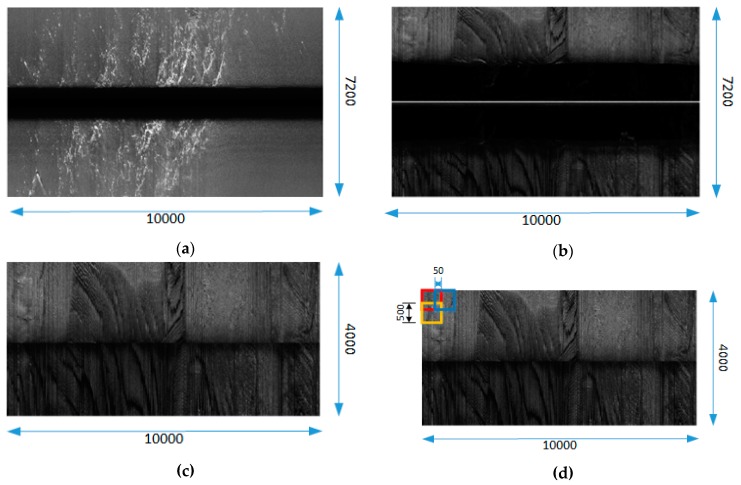
The high-resolution sonar image process. (**a**) The coral reef image at high-resolution; (**b**) The sand wave image at high-resolution; (**c**) sonar image after removing water column; (**d**) patch-wise strategy.

**Figure 10 sensors-19-01985-f010:**
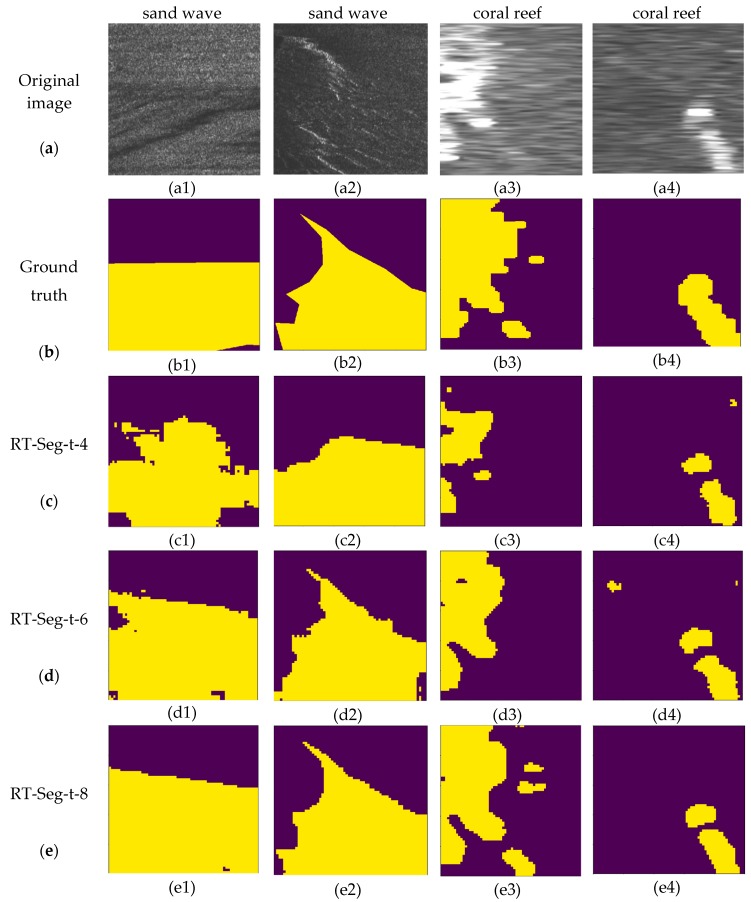
The segmentation results of SSS images, in which the first two columns are sand wave images from Dataset-1, the last two are coral reef images from Dataset-2. (**a**) original SSS image with a size of 500 × 500; (**b**) ground truth; (**c**–**e**) the segmentation results of RT-Seg-t-4, RT-Seg-t-6 and RT-Seg-t-8, respectively.

**Figure 11 sensors-19-01985-f011:**
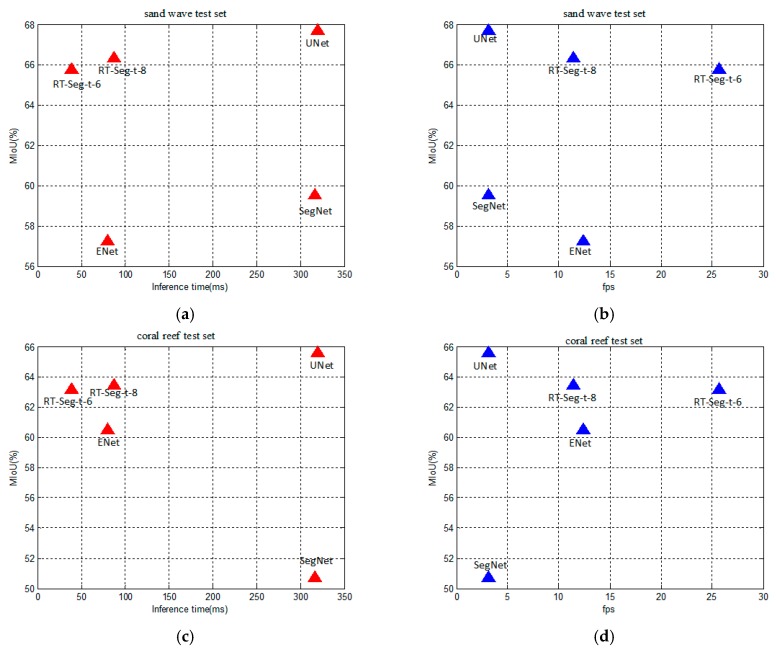
The trade-off between accuracy and speed. (**a**) The trade-off between the inference time for a single input frame and accuracy of different networks in sand wave test set; (**b**) The trade-off between the number of frames that can be processed per second and accuracy of different networks in sand wave test set; **(c**) The trade-off between the inference time for a single input frame and accuracy of different networks in coral reef test set; (**d**) The trade-off between the number of frames that can be processed per second and accuracy of different networks in coral reef test set.

**Figure 12 sensors-19-01985-f012:**
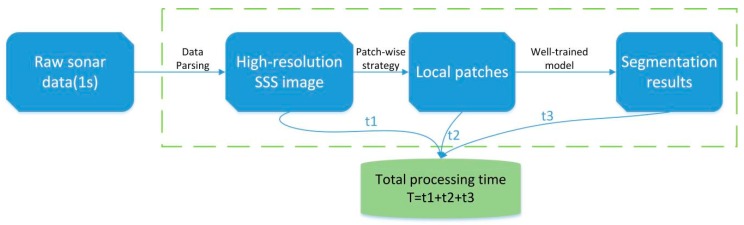
The real-time test processing experiment.

**Table 1 sensors-19-01985-t001:** The encoder block implementation.

Branch	Input	Operator	Output
	h×w×k	×1, conv, BN, ReLU	h×w×(tk)
Left branch	h×w×(tk)	3 × 3, DWconv, BN, ReLU	h×w×(tk)
	h×w×(tk)	1 × 1, PWconv, BN, Linear	h×w×k
Right branch	h×w×(tk)	5 × 5, DWconv, BN, ReLU	h×w×(tk)
	h×w×(tk)	1 × 1, PWconv, BN, Linear	h×w×k
	h×w×k	Filter Concatenate	h×w×(2k)
	h×w×(2k)	Max-pooling	(h/2)×(w/2)×(2k)

**Table 2 sensors-19-01985-t002:** Quantitative comparison on sand wave test set (The best result is bold).

Network	PA	MPA	MIOU	FWIOU
SegNet	90.09	80.77	59.54	83.24
UNet	**92.79**	**83.11**	**67.70**	**85.77**
ENet	89.23	79.12	57.23	82.11
RT-Seg-t-4	86.53	76.24	53.46	79.36
RT-Seg-t-6	91.33	82.32	65.78	83.46
RT-Seg-t-8	91.42	82.13	66.32	86.38

**Table 3 sensors-19-01985-t003:** Quantitative comparison on coral reef test set (The best result is bold).

Network	PA	MPA	MIOU	FWIOU
SegNet	91.56	83.76	50.72	84.57
UNet	**93.88**	**85.62**	**66.62**	**88.12**
ENet	90.10	80.15	60.48	81.59
RT-Seg-t-4	85.64	85.93	56.82	85.15
RT-Seg-t-6	92.43	84.32	63.17	86.10
RT-Seg-t-8	92.67	85.13	63.45	86.97

**Table 4 sensors-19-01985-t004:** Processing time comparison (The best result is bold).

Model	NVIDIA Jetson AGX Xavier			
	224 × 224		500 × 500	
	ms	fps	Ms	fps
SegNet	72.9	13.717	316.2	3.162
UNet	68.2	14.662	320.1	3.124
ENet	54.4	18.382	80.5	12.422
RT-Seg-t-6	**25.6**	**39.063**	**38.9**	**25.678**
RT-Seg-t-8	60.4	16.547	87.4	11.435

**Table 5 sensors-19-01985-t005:** Hardware requirements (The best result is bold).

Network	GFLOPs	Parameters	Model Size (fp16)
SegNet	286.0	29.5M	58.9M
UNet	328.1	31.03M	62.04M
ENet	3.83	**0.35M**	**0.7M**
RT-Seg-t-6	**2.14**	0.46M	1.4M
RT-Seg-t-8	4.96	1.4M	2.5M

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
