# Peer review of "RT-Seg: A Real-Time Semantic Segmentation Network for Side-Scan Sonar Images"

_sensors, 2019, doi:10.3390/s19091985_

Round 1
Reviewer 1 Report
The goal of the paper was to present a novel segmentation architecture using neural network that allows real-time processing of high resolution Side-Scan Sonar images. It is vital especially for the developing an application to sonar searches/surveys using autonomous underwater vehicle (AUV). The importance of a trade-off between accuracy and speed of segmentation process is crucial. The authors inspiration was SegNet, an encoder-decoder structure that is widely applied for semantic segmentation. Authors proposed using depth-wise separable convolution in the encoder part and patch-wise strategy to split high resolution sonar image into local patches for network training. Presented approach was then tested and compared with existing network architectures.
The structure of the paper is clear. The sections are in a logical order. In my opinion (as non-native English speaker), the language and style are correct which make reception of the paper very easy and pleasant. Background research on the subject is thorough. The authors clearly explain their choices on using segmentation networks and depth-wise convolution to solve sonar image segmentation problem in RT . Proposed architecture is described in details, backed with clear schemas (fig 3-7). Experimental process is also presented together with sonar data pre-processing and some data samples (fig 9., fig. 4). Obtained results are analyzed in terms of quality and quantity. The method as a whole is well-thought and the experiment shows clearly the advantage of proposed approached against other solutions ( not dedicated for SSS images SegNet, UNet, ENet).
Two minor remarks:
Page 1 line 41: high resolution is definitely not a general SSS characteristic – it depends on sonars frequency ( and therefore range), which is very different for different application ( marine, inland, shallow water etc.)
Conclusions: how authors see further development and research on segmentation of SSS images in RT?
To sum up: The paper is a solid work on application of CNN for processing SSS images. The results are promising, especially in relation to applicability in AUV systems for improving their intelligence and autonomy of navigation/surveying.
Author Response
Dear Reviewer:
Thank you very much for your careful review and constructive suggestions with regard to our manuscript "RT-Seg: A Real-Time Semantic Segmentation Network for Side-Scan Sonar images".Those comments are all valuable and very helpful for revising and improving our paper, as well as the importance guiding significance to our researches.We have studied comments carefully and have made correction which we hope meet with approval.Our main corrections in the manuscript and the responds to the reviewer's comments are a PDF file "Response to Reviewer 1 Comments.pdf".Revised portion are marked in red in the paper. Thank you very much for your comments and suggestions once again.
Thank you and best regards!
Yours sincerely,
Qi Wang , Meihan Wu , Fei Yu , Chen Feng * , kaige Li , Yuemei Zhu , Eric Rigall , Bo He

Reviewer 2 Report
Please see the attached.

Author Response
Dear Reviewer:
Thank you very much for your careful review and constructive suggestions with regard to our manuscript "RT-Seg: A Real-Time Semantic Segmentation Network for Side-Scan Sonar images".Those comments are all valuable and very helpful for revising and improving our paper, as well as the importance guiding significance to our researches.We have studied comments carefully and have made correction which we hope meet with approval.Our main corrections in the manuscript and the responds to the reviewer's comments are a PDF file "Response to Reviewer 2 Comments.pdf".Revised portion are marked in red in the paper. Thank you very much for your comments and suggestions once again.
Thank you and best regards!
Yours sincerely,
Qi Wang , Meihan Wu , Fei Yu , Chen Feng * , kaige Li , Yuemei Zhu , Eric Rigall , Bo He
